# LEARNING-AUGMENTED SKETCHES FOR HESSIANS

## ABSTRACT

Sketching is a dimensionality reduction technique where one compresses a matrix by often random linear combinations. A line of work has shown how to sketch the Hessian to speed up each iteration in a second order method, but such sketches usually depend only on the matrix at hand, and in a number of cases are even oblivious to the input matrix. One could instead hope to learn a distribution on sketching matrices that is optimized for the specific distribution of input matrices. We show how to design learned sketches for the Hessian in the context of second order methods, where we learn potentially different sketches for the different iterations of an optimization procedure. We show empirically that learned sketches, compared with their "non-learned" counterparts, improve the approximation accuracy for important problems, including LASSO, SVM, and matrix estimation with nuclear norm constraints. Several of our schemes can be proven to perform no worse than their unlearned counterparts.

## 1 INTRODUCTION

Large-scale optimization problems are abundant and solving them efficiently requires powerful tools to make the computation practical. This is especially true of second order methods which often are less practical than first order ones. Although second order methods may have many fewer iterations, each iteration could involve inverting a large Hessian, which is cubic time; in contrast, first order methods such as stochastic gradient descent are linear time per iteration.

In order to make second order methods faster in each iteration, a large body of work has looked at dimensionality reduction techniques, such as sampling, sketching, or approximating the Hessian by a low rank matrix. See, for example, (Gower et al., 2016; Xu et al., 2016; Pilanci & Wainwright, 2016; 2017; Doikov & Richtárik, 2018; Gower et al., 2018; Roosta-Khorasani & Mahoney, 2019; Gower et al., 2019; Kylasa et al., 2019; Xu et al., 2020; Li et al., 2020). Our focus is on sketching techniques, which often consist of multiplying the Hessian by a random matrix chosen independently of the Hessian. Sketching has a long history in theoretical computer science (see, e.g., (Woodruff, 2014) for a survey), and we describe such methods more below. A special case of sketching is sampling, which in practice is often uniform sampling, and hence oblivious to properties of the actual matrix. Other times the sampling is non-uniform, and based on squared norms of submatrices of the Hessian or on the so-called leverage scores of the Hessian.

Our focus is on sketching techniques, and in particular, we follow the framework of (Pilanci & Wainwright, 2016; 2017) which introduce the iterative Hessian sketch and Newton sketch, as well as the high accuracy refinement given in (van den Brand et al., 2020). If one were to run Newton's method to find a point where the gradient is zero, in each iteration one needs to solve an equation involving the current Hessian and gradient to find the update direction. When the Hessian can be decomposed as $A^\top A$ for an $n \times d$ matrix $A$ with $n \gg d$, then sketching is particularly suitable. The *iterative Hessian sketch* was proposed in (Pilanci & Wainwright, 2016), where $A$ is replaced with $S \cdot A$, for a random matrix $S$ which could be i.i.d. Gaussian or drawn from a more structured family of random matrices such as the Subsampled Randomized Hadamard Transforms or COUNT-SKETCH matrices; the latter was done in (Cormode & Dickens, 2019). The *Newton sketch* was proposed in (Pilanci & Wainwright, 2017), which extended sketching methods beyond constrained least-squares problems to any twice differentiable function subject to a closed convex constraint set. Using this sketch inside of interior point updates has led to much faster algorithms for an extensive body of convex optimization problems Pilanci & Wainwright (2017). By instead using sketching as

a preconditioner, an application of the work of (van den Brand et al., 2020) (see Appendix E) was able to improve the dependence on the accuracy parameter $\epsilon$ to logarithmic.

In general, the idea behind sketching is the following. One chooses a random matrix $S$, drawn from a certain family of random matrices, and computes $SA$. If $A$ is tall-and-skinny, then $S$ is short-and-fat, and thus $SA$ is a small, roughly square matrix. Moreover, $SA$ preserves important properties of $A$. One typically desired property is that $S$ is a subspace embedding, meaning that simultaneously for all $x$, one has $\|SAx\|_2 = (1 \pm \epsilon)\|Ax\|_2$. An observation exploited in (Cormode & Dickens, 2019), building off of the COUNT-SKETCH random matrices $S$ introduced in randomized linear algebra in (Clarkson & Woodruff, 2017), is that if $S$ contains $O(1)$ non-zero entries per column, then $SA$ can be computed in $O(\text{nnz}(A))$ time, where $\text{nnz}(A)$ denotes the number of nonzeros in $A$. This is sometimes referred to as input sparsity running time.

Each iteration of a second order method often involves solving an equation of the form $A^\top A x = A^\top b$, where $A^\top A$ is the Hessian and $b$ is the gradient. For a number of problems, one has access to a matrix $A \in \mathbb{R}^{n \times d}$ with $n \gg d$, which is also an assumption made in Pilanci & Wainwright (2017). Therefore, the solution $x$ is the minimizer to a constrained least squares regression problem:

$$\min_{x \in \mathcal{C}} \frac{1}{2}\|Ax - b\|_2^2, \tag{1}$$

where $\mathcal{C}$ is a convex constraint set in $\mathbb{R}^d$. For the unconstrained case ($\mathcal{C} = \mathbb{R}^d$), various classical sketches that attain the subspace embedding property can provably yield high-accuracy approximate solutions (see, e.g., (Sarlos, 2006; Nelson & Nguyên, 2013; Cohen, 2016; Clarkson & Woodruff, 2017)); for the general constrained case, the Iterative Hessian Sketch (IHS) was proposed by Pilanci & Wainwright (2016) as an effective approach and Cormode & Dickens (2019) employed sparse sketches to achieve input-sparsity running time for IHS. All sketches used in these results are data-oblivious random sketches.

**Learned Sketching.** In the last few years, an exciting new notion of *learned sketching* has emerged. Here the idea is that one often sees independent samples of matrices $A$ from a distribution $\mathcal{D}$, and can train a model to learn the entries in a sketching matrix $S$ on these samples. When given a future sample $B$, also drawn from $\mathcal{D}$, the learned sketching matrix $S$ will be such that $S \cdot B$ is a much more accurate compression of $B$ than if $S$ had the same number of rows and were instead drawn without knowledge of $\mathcal{D}$. Moreover, the learned sketch $S$ is often *sparse*, therefore allowing $S \cdot B$ to be applied very quickly. For large datasets $B$ this is particularly important, and distinguishes this approach from other transfer learning approaches, e.g., (Andrychowicz et al., 2016), which can be considerably slower in this context.

Learned sketches were first used in the data stream context for finding frequent items (Hsu et al., 2019) and have subsequently been applied to a number of other problems on large data. For example, Indyk et al. (2019) showed that learned sketches yield significantly small errors for low rank approximation. In (Dong et al., 2020), significant improvements to nearest neighbor search were obtained via learned sketches. More recently, Liu et al. (2020) extended learned sketches to several problems in numerical linear algebra, including least-squares and robust regression, as well as $k$-means clustering.

Despite the number of problems that learned sketches have been applied to, they have not been applied to convex optimization in general. Given that such methods often require solving a large overdetermined least squares problem in each iteration, it is hopeful that one can improve each iteration using learned sketches. However, a number of natural questions arise: (1) *how should we learn the sketch?* (2) *should we apply the same learned sketch in each iteration, or learn it in the next iteration by training on a data set involving previously learned sketches from prior iterations?*

**Our Contributions.** In this work we answer the above questions and derive the first learned sketches for a wide number of problems in convex optimization.

Namely, we apply learned sketches to constrained least-squares problems, including LASSO, support vector machines (SVM), and matrix regression with nuclear norm constraints. We show empirically that learned sketches demonstrate superior accuracy over random oblivious sketches for each of these problems. Specifically, compared with three classical sketches (Gaussian, COUNT-SKETCH and Sparse Johnson-Lindenstrauss Transforms; see definitions in Section 2), the learned sketches in each of the first few iterations

- improve the LASSO error $f(x) - f(x^*)$ by 80% to 87% in two real-world datasets, where $f(x) = \frac{1}{2} \|Ax - b\|_2^2 + \|x\|_1$;
- improve the dual SVM error $f(x) - f(x^*)$ by 10–30% for a synthetic and a real-world dataset, as well as by 30%–40% for another real-world dataset, where $f(x) = \|Bx\|_2^2$;
- improve the matrix estimation error $f(X) - f(X^*)$ by at least 30% for a synthetic dataset and at least 95% for a real-world data set, where $f(X) = \|AX - B\|_F^2$.

Therefore, the learned sketches attain a smaller error within the same number of iterations, and in fact, within the same limit on the maximum runtime, since our sketches are extremely sparse (see below).

We also study the general framework of convex optimization in (van den Brand et al., 2020), and show that also for sketching-based preconditioning, learned sketches demonstrate considerable advantages. More precisely, by using a learned sketch with the same number of rows as an oblivious sketch, we are able to obtain a much better preconditioner with the same overall running time.

All of our learned sketches $S$ are extremely sparse, meaning that they contain a single non-zero entry per column. Following the previous work of (Indyk et al., 2019), we choose the position of the nonzero entry in each column to be uniformly random, while the value of the nonzero entry is learned. This already demonstrates a significant advantage over non-learned sketches, and has a fast training time. Importantly, because of such sparsity, our sketches can be applied in input sparsity time given a new optimization problem.

We also provide several theoretical results, showing how to algorithmically use learned sketches in conjunction with random sketches so as to do no worse than random sketches.

## 2 PRELIMINARIES

**Classical Sketches.** Below we review several classical sketches that have been used for solving optimization problems.

- Gaussian sketch: $S = \frac{1}{\sqrt{m}} G$, where $G$ is an $m \times n$ Gaussian random matrix.
- COUNT-SKETCH: Each column of $S$ has only a single non-zero entry. The position of the non-zero entry is chosen uniformly over the $m$ entries in the column and the value of the entry is either $+1$ or $-1$, each with probability $1/2$. Further, the columns are chosen independently.
- Sparse Johnson-Lindenstrauss Transform (SJLT): $S$ is the vertical concatenation of $s$ independent COUNT-SKETCH matrices, each of dimension $m/s \times n$.

**COUNT-SKETCH-type Sketch.** A COUNT-SKETCH-type sketch is characterized by a tuple $(m, n, p, v)$, where $m, n$ are positive integers and $p, v$ are $n$-dimensional real vectors, defined as follows. The sketching matrix $S$ has dimensions $m \times n$ and $S_{p_i, i} = v_i$ for all $1 \le i \le n$ while all the other entries of $S$ are 0. When $m$ and $n$ are clear from context, we may characterize such a sketching matrix by $(p, v)$ only.

**Subspace Embeddings.** For a matrix $A \in \mathbb{R}^{n \times d}$, we say a matrix $S \in \mathbb{R}^{m \times n}$ is a $(1 \pm \epsilon)$-subspace embedding for the column span of $A$ if $(1 - \epsilon) \|Ax\|_2 \le \|SAx\|_2 \le (1 + \epsilon) \|Ax\|_2$ for all $x \in \mathbb{R}^d$. The classical sketches above, with appropriate parameters, are all subspace embedding matrices with at least a constant probability; our focus is on COUNT-SKETCH which can be applied in input sparsity running time. We summarize the parameters needed for a subspace embedding below:

- Gaussian sketch: $m = O(d/\epsilon^2)$. It is a dense matrix and computing $SA$ costs $O(m \cdot \mathrm{nnz}(A)) = O(\mathrm{nnz}(A)d/\epsilon^2)$ time.
- COUNT-SKETCH: $m = O(d^2/\epsilon^2)$ (Clarkson & Woodruff, 2017). Although the number of rows is quadratic in $d/\epsilon$, the sketch matrix $S$ is sparse and computing $SA$ takes only $O(\mathrm{nnz}(A))$ time.
- SJLT: $m = O(d/\epsilon^2)$ and has $s = O(1/\epsilon)$ non-zeros per column (Nelson & Nguyên, 2013; Cohen, 2016). Computing $SA$ takes $O(s \, \mathrm{nnz}(A)) = O(\mathrm{nnz}(A)/\epsilon)$ time.

**Iterative Hessian Sketch.** The Iterative Hessian Sketching (IHS) method (Pilanci & Wainwright, 2016) solves the constrained least-squares problem (1) by iteratively performing the update

$$x_{t+1} = \arg\min_{x \in \mathcal{C}} \frac{1}{2} \|S_{t+1} A(x - x_t)\|_2^2 - \langle A^\top (b - Ax_t), x - x_t \rangle, \tag{2}$$

where $S_{t+1}$ is a sketching matrix. It is not difficult to see that for the unsketched version ($S_{t+1}$ is the identity matrix) of the minimization above, the optimal solution $x^{t+1}$ coincides with the optimal solution to the constrained least square problem (1). The IHS approximates the Hessian $A^\top A$ by a sketched version $(S_{t+1}A)^\top(S_{t+1}A)$ to improve runtime, as $S_{t+1}A$ typically has very few rows.

**Unconstrained Convex Optimization.** Consider an unconstrained convex optimization problem $\min_x f(x)$, where $f$ is smooth and strongly convex, and its Hessian $\nabla^2 f$ is Lipschitz continuous. This problem can be solved by Newton's method, which iteratively performs the update

$$x_{t+1} = x_t - \arg\min_z \left\| (\nabla^2 f(x_t)^{1/2})^\top (\nabla^2 f(x_t)^{1/2})z - \nabla f(x_t) \right\|_2,$$

provided it is given a good initial point $x_0$. In each step, it requires solving a regression problem of the form $\min_z \left\| A^\top A z - y \right\|_2$, which, with access to $A$, can be solved with a fast regression solver in (van den Brand et al., 2020). The regression solver first computes a preconditioner $R$ via a QR decomposition such that $SAR$ has orthonormal columns, where $S$ is a sketching matrix, then solves $\widehat{z} = \arg\min_{z'} \left\| (AR)^\top (AR)z' - y \right\|_2$ by gradient descent and returns $R\widehat{z}$ in the end. Here, the point of sketching is that the QR decomposition of $SA$ can be computed much more efficiently than the QR decomposition of $A$ since $S$ has only a small number of rows.

**Learning a Sketch.** We use the same learning algorithm in (Liu et al., 2020), given in Algorithm 1. The algorithm aims to minimize the mean loss function $\mathcal{L}(S, \mathcal{A}) = \frac{1}{N}\sum_{i=1}^{N} \mathcal{L}(S, A_i)$, where $S$ is the learned sketch, $\mathcal{L}(S, A)$ is the loss function of $S$ applied to a data matrix $A$, and $\mathcal{A} = \{A_1, \ldots, A_N\}$ is a (random) subset of training data.

## 3  HESSIAN SKETCH

In this section, we consider the minimization problem

$$\min_{x \in \mathcal{C}} \left\{ \frac{1}{2} \|SAx\|_2^2 - \langle A^\top y, x \rangle \right\}, \quad (3)$$

which is used as a subroutine for the IHS (cf. (2)). We present an algorithm with the learned sketch in Algorithm 2. To analyze its performance, we let $\mathcal{R}$ be the column space of $A \in \mathbb{R}^{m \times n}$ and define the following quantities (corresponding exactly to the unconstrained case in Pilanci & Wainwright (2016))

$$Z_1(S) = \inf_{v \in \mathcal{R} \cap \mathbb{S}^{n-1}} \|Sv\|_2^2,$$

$$Z_2(S) = \sup_{u,v \in \mathcal{R} \cap \mathbb{S}^{n-1}} \left\langle u, (S^\top S - I_n)v \right\rangle,$$

where $\mathbb{S}^{n-1}$ denotes the Euclidean unit sphere in $\mathbb{R}^n$.

The following is the estimation guarantee of $\widehat{Z}_1$ and $\widehat{Z}_2$. The proof is postponed to Appendix A.

---

**Algorithm 1** LEARN-SKETCH: Gradient descent algorithm for learning the sketch values

**Require:** $\mathcal{A}_{train} = \{A_1, ..., A_N\}$ ($A_i \in \mathbb{R}^{n \times d}$), learning rate $\alpha$
1: Randomly initialize $p, v$ for a Count-Sketch-type sketch
2: **for** $t = 0$ **to** $steps$ **do**
3:     Form $S$ using $p, v$
4:     Sample batch $\mathcal{A}_{batch}$ from $\mathcal{A}_{train}$
5:     $v \leftarrow v - \alpha \frac{\partial \mathcal{L}(S, \mathcal{A}_{batch})}{\partial v}$
6: **end for**

---

**Algorithm 2** Solver for (3)

1: $S_1 \leftarrow$ learned sketch, $S_2 \leftarrow$ random sketch
2: $(\widehat{Z}_{i,1}, \widehat{Z}_{i,2}) \leftarrow$ ESTIMATE$(S_i, A)$, $i = 1, 2$
3: **if** $\widehat{Z}_{1,2}/\widehat{Z}_{1,1} < \widehat{Z}_{2,2}/\widehat{Z}_{2,1}$ **then**
4:     $\widehat{x} \leftarrow$ solution of (3) with $S = S_1$
5: **else**
6:     $\widehat{x} \leftarrow$ solution of (3) with $S = S_2$
7: **end if**
8: **return** $\widehat{x}$
9: **function** ESTIMATE$(S, A)$
10:     $T \leftarrow$ sparse $(1 \pm \eta)$-subspace embedding matrix for $d$-dimensional subspaces
11:     $(Q, R) \leftarrow$ QR$(TA)$
12:     $\widehat{Z}_1 \leftarrow \sigma_{\min}(SAR^{-1})$
13:     $\widehat{Z}_2 \leftarrow (1 \pm \eta)$-approximation to $\left\| (SAR^{-1})^\top (SAR^{-1}) - I \right\|_{op}$
14:     **return** $(\widehat{Z}_1, \widehat{Z}_2)$
15: **end function**

---

**Lemma 3.1.** *Suppose that $\eta \in (0, \frac{1}{3})$ is a small constant, A is of full rank and $S$ has $O(d^2)$ rows. The function ESTIMATE$(S, A)$ returns in $O((\mathrm{nnz}(A)\log(1/\eta) + \mathrm{poly}(d/\eta))$ time $\widehat{Z}_1, \widehat{Z}_2$ which with probability at least $0.99$ satisfy that $\frac{Z_1(S)}{(1+\eta)} \leq \widehat{Z}_1 \leq \frac{Z_1(S)}{1-\eta}$ and $\frac{Z_2(S)}{(1+\eta)^2} - 3\eta \leq \widehat{Z}_2 \leq \frac{Z_2(S)}{(1-\eta)^2} + 3\eta$.*

Note for a matrix $A$, $\|A\|_{op} = \sup_{x \neq 0} \frac{\|Ax\|_2}{\|x\|_2}$ is its operator norm. Similar to (Pilanci & Wainwright, 2016, Proposition 1), we have the following guarantee. The proof is postponed to Appendix B.

**Theorem 3.2.** *Let $\eta \in (0, \frac{1}{3})$ be a small constant. Suppose that A is of full rank and $S_1$ and $S_2$ are both* COUNT-SKETCH-*type sketches with $O(d^2)$ rows. Algorithm 2 returns a solution $\widehat{x}$ which, with probability at least 0.98, satisfies that $\|A(\widehat{x} - x^*)\|_2 \leq (1+\eta)^4 \left( \min \left\{ \frac{\widehat{Z}_{1,2}}{\widehat{Z}_{1,1}}, \frac{\widehat{Z}_{2,2}}{\widehat{Z}_{2,1}} \right\} + 4\eta \right) \|Ax^*\|_2$ in $O(\mathrm{nnz}(A) \log(\frac{1}{\eta}) + \mathrm{poly}(\frac{d}{\eta}))$ time, where $x^* = \arg\min_{x \in \mathcal{C}} \|Ax - b\|_2$ is the least-squares solution.*

## 4 HESSIAN REGRESSION

In this section, we consider the minimization problem

$$\min_z \|A^\top A z - y\|_2, \qquad (4)$$

which is used as a subroutine for the unconstrained convex optimization problem $\min_x f(x)$ with $A^\top A$ being the Hessian matrix $\nabla^2 f(x)$ (See Section 2). Here $A \in \mathbb{R}^{n \times d}$, $y \in \mathbb{R}^d$, and we have access to $A$. We incorporate a learned sketch into the fast regression solver in (van den Brand et al., 2020) and present the algorithm in Algorithm 3.

---

**Algorithm 3** Fast Regression Solver for (4)

1: $S_1 \leftarrow$ learned sketch, $S_2 \leftarrow$ random sketch
2: $(Q_i, R_i) \leftarrow \mathrm{QR}(S_i A)$, $i = 1, 2$
3: $(\sigma_i, \sigma_i') \leftarrow \mathrm{EIG}(AR_i^{-1})$, $i = 1, 2$
        $\triangleright$ $\mathrm{EIG}(B)$ returns the max and min singular values of $B$
4: **if** $\sigma_1/\sigma_1' < \sigma_2/\sigma_2'$ **then**
5:     $P \leftarrow R_1^{-1}$, $\eta \leftarrow 1/(\sigma_1^2 + (\sigma_1')^2)$
6: **else**
7:     $P \leftarrow R_2^{-1}$, $\eta \leftarrow 1/(\sigma_2^2 + (\sigma_2')^2)$
8: **end if**
9: $z_0 \leftarrow 0$
10: **while** $\|A^\top A P z_t - y\|_2 \geq \epsilon \|y\|_2$ **do**
11:     $z_{t+1} \leftarrow z_t - \eta(P^\top A^\top A P)(P^\top A^\top A P z_t - P^\top y)$
12: **end while**
13: **return** $P z_t$

---

Here the subroutine $\mathrm{EIG}(B)$ applies a $(1 + \eta)$-subspace embedding sketch $T$ to $B$ for some small constant $\eta$ and returns the maximum and the minimum singular values of $TB$. Since $B$ admits the form of $AR$, the sketched matrix $TB$ can be calculated as $(TA)R$ and thus can be computed in $O(\mathrm{nnz}(A) + \mathrm{poly}(d))$ time if $T$ is a COUNT-SKETCH matrix of $O(d^2)$ rows. The extreme singular values of $TB$ can be found by SVD or the Lanczos's algorithm.

Similar to Lemma 4.2 in (van den Brand et al., 2020), we have the following guarantee of Algorithm 3. The proof parallels the proof in (van den Brand et al., 2020) and is postponed to Appendix C.

**Theorem 4.1.** *Suppose that $S_1$ and $S_2$ are both* COUNT-SKETCH-*type sketches with $O(d^2)$ rows. Algorithm 3 returns a solution $x'$ such that $\|A^\top A x' - y\|_2 \leq \epsilon \|y\|_2$ in $O(\mathrm{nnz}(A)) + \widetilde{O}(nd \cdot (\min\{\sigma_1/\sigma_1', \sigma_2/\sigma_2'\})^2 \cdot \log(\kappa(A)/\epsilon) + \mathrm{poly}(d))$ time.*

**Remark 4.2.** In Algorithm 3, $S_2$ can be chosen to be a subspace embedding matrix for $d$-dimensional subspaces, in which case, $AR_2^{-1}$ has condition number close to 1 (see, e.g., (Woodruff, 2014, p38)) and the full algorithm would run faster than the trivial $O(nd^2)$-time solver to (4).

**Remark 4.3.** For the original unconstrained convex optimization problem $\min_x f(x)$, one can run the entire optimization procedure with learned sketches versus the entire optimization procedure with random sketches, compare the objective values at the end, and choose the better of the two. For least-squares $f(x) = \frac{1}{2} \|Ax - b\|_2^2$, the value of $f(x)$ can be approximated efficiently by a sparse subspace embedding matrix in $O(\mathrm{nnz}(A) + \mathrm{nnz}(b) + \mathrm{poly}(d))$ time.

## 5 IHS EXPERIMENTS

**Training.** We learn the sketching matrix in each iteration (also called a round) separately. Recall that in the $(t+1)$-st round of IHS, the optimization problem we need to solve (2) depends on $x_t$. An issue is that we do not know $x_t$, which is needed to generate the training data for the $(t+1)$-st round. Our solution is to use the sketch matrix in the previous round to obtain $x_t$ by solving (2), and then use it to generate the training data in the next round. That is, in the first round, we train the sketch matrix $S_1$ to solve the problem $x_1 = \arg\min_{x \in \mathcal{C}} \frac{1}{2} \|S_1 A x\|_2^2 - \langle A^\top b, x \rangle$ and use $x_1$ to generate the training data for the optimization problem for $x_2$, and so on. The loss function we use is the unsketched objective function in the $(t+1)$-st iteration, i.e., $\mathcal{L}(S_{t+1}, A) = \frac{1}{2} \|A(x_{t+1} - x_t)\|_2^2 - \langle A^T(b - Ax_t), x_{t+1} - x_t \rangle$, where $x_{t+1}$ is the solution to (2) and thus depends on $S_{t+1}$.

**Comparison.** We compare the learned sketch against three classical sketches: Gaussian, COUNT-SKETCH, and SJLT (see Section 2) in all experiments. The quantity we compare is a certain error, defined individually for each problem, in each round of the iteration of the IHS or as a function of the runtime of the algorithm. All of our experiments are conducted on a laptop with 1.90GHz CPU and 16GB RAM.

## 5.1 LASSO

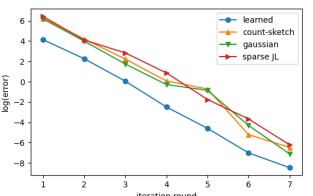

Figure 1: Test error of LASSO on CO emissions dataset, $m = 5d$

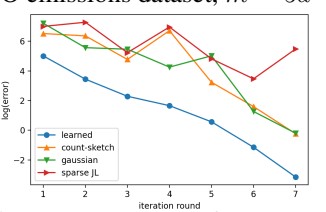

Figure 2: Test error of LASSO on CO emissions dataset, $m = 3d$

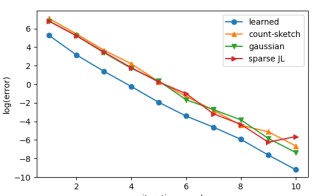

Figure 3: Test error of LASSO on greenhouse gas dataset, $m = 6d$

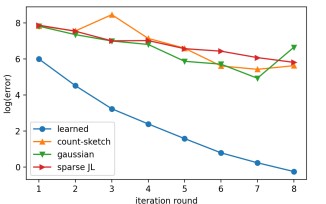

Figure 4: Test error of LASSO on greenhouse gas dataset, $m = 3.5d$

We define an instance of LASSO regression as

$$x^* = \arg\min_{x \in \mathbb{R}^d} \frac{1}{2} \|Ax - b\|_2^2 + \lambda \|x\|_1 , \tag{5}$$

where $\lambda$ is a parameter. We use two real-world datasets:

- **CO emission**[1]: the dataset contains 9 sensor measures aggregated over one hour (by means of average or sum), which can help to predict the CO emission. We divide the raw data into 120 $(A_i, b_i)$ such that $A_i \in \mathbb{R}^{300 \times 9}$, $b_i \in \mathbb{R}^{300 \times 1}$. The data in each matrix is sorted in chronological order. $|(A, b)_{\text{train}}| = 96$, $|(A, b)_{\text{test}}| = 24$.
- **Greenhouse gas**[2]: time series of measured greenhouse gas concentrations in the California atmosphere. Each $(A, b)$ corresponds to a different measurement location. $A_i \in \mathbb{R}^{327 \times 14}$, $b_i \in \mathbb{R}^{327 \times 1}$, and $|(A, b)_{\text{train}}| = 400$, $|(A, b)_{\text{test}}| = 100$. (This dataset was also used in (Liu et al., 2020).)

We choose $\lambda = 1$ in the LASSO regression (5). We choose $m = 5d$ and $m = 3d$ for the CO emission dataset and $m = 6d$ and $m = 3.5d$ for the greenhouse gas dataset. We consider the error $(\frac{1}{2} \|Ax - b\|_2^2 + \|x\|_1) - (\frac{1}{2} \|Ax^* - b\|_2^2 + \|x^*\|_1)$ and take an average over five independent trials. We plot in (natural) logarithmic scale the mean errors of the two datasets in Figures 1 to 4. We can observe that the learned sketches consistently outperform the classical random sketches in all cases. Note that our sketch size is much smaller than the theoretical bound. For a smaller sketch size, classical random sketches converge slowly or do not yield stable results, while the learned sketches can converge to a small error quickly. For larger sketch sizes, all three classical sketches have approximately the same order of error, and the learned sketches reduce the error by a $5/6$ to $7/8$ factor in all iterations.

## 5.2 SUPPORT VECTOR MACHINE

In the context of binary classification, a labeled sample is a pair $(a_i, z_i)$, where $a_i \in \mathbb{R}^n$ is a vector representing a collection of features and $z_i \in \{-1, +1\}$ is the associated class label. Given a set of labeled patterns $\{(a_i, z_i)\}_{i=1}^d$, the support vector machine (SVM) estimates the weight vector $w^*$ by minimizing the function

$$w^* = \arg\min_{w \in \mathbb{R}^n} \left\{ \frac{C}{2} \sum_{i=1}^d g(z_i, \langle w_i, a_i \rangle) + \frac{1}{2} \|w\|_2^2 \right\},$$

where $C$ is a parameter. Here we use the squared hinge loss $g(z_i, \langle w, a_i \rangle) := (1 - z_i \langle w, a_i \rangle)_+^2$.

---

[1] https://archive.ics.uci.edu/ml/datasets/Gas+Turbine+CO+and+NOx+Emission+Data+Set
[2] https://archive.ics.uci.edu/ml/datasets/Greenhouse+Gas+Observing+Network

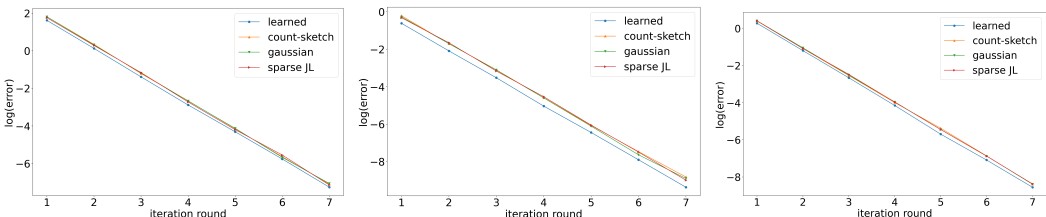

Figure 5: Test error of SVM on random Gaussian dataset

Figure 6: Test error of SVM on swarm behavior dataset

Figure 7: Test error of SVM on Gisette dataset

The dual of this problem can be written as a constrained minimization problem (see, e.g., (Li et al., 2009; Pilanci & Wainwright, 2015)),

$$x^* := \arg\min_{x \in \Delta^d} \|Bx\|_2^2 \,,$$

over the domain $\Delta^d = \{x \in \mathbb{R}^d : x \geq 0 \text{ and } \|x\|_1 = 1\}$, the positive simplex in $\mathbb{R}^d$. Here $B = [(AD)^\top \frac{1}{\sqrt{C}} I_d]^\top \in \mathbb{R}^{(n+d) \times d}$, where $A$ is an $n \times d$ matrix with $a_i \in \mathbb{R}^n$ as its $i$-th column and $D = \mathrm{diag}(z)$ is a $d \times d$ diagonal matrix.

We conduct experiments on the following three datasets:

- **Random Gaussian** (synthetic): We follow the same construction in Pilanci & Wainwright (2016). We generate a two-component Gaussian mixture model, based on the component distributions $N(\mu_0, I)$ and $N(\mu_1, I)$, where $\mu_0$ and $\mu_1$ are uniformly distributed in $[-3, 3]$. Placing equal weights on each component, we draw $d$ samples from this mixture distribution.
- **Swarm behavior**[3]: Each instance in the dataset has $n = 2400$ features and the task is to predict whether the instance is flocking or not flocking. We use only the first 6000 instances of the raw data, and divide them into 200 smaller groups of instances. Each group contains $d = 30$ instances, corresponding to a $B_i$ of size $2430 \times 30$. The training data consists of 160 groups and the test data consist of 40 groups.
- **Gisette**[4]: Gisette is a handwritten digit recognition problem which asks to to separate the highly confusable digits '4' and '9'. Each instance has $n = 5000$ features. The raw data is divided into 200 smaller groups, where each contains $d = 30$ instances and corresponds to a $B_i$ of size $5030 \times 30$. The training data consists of 160 groups and the test data consists of 40 groups.

We choose $m = 10d$ in all experiments and define the error as $\|Bx\|_2^2 - \|Bx^*\|_2^2$. For random sketches, we take the average error over five independent trials, and for learned sketches, over three independent trials. For the Gisette dataset, we use the learned sketch in all rounds. We plot in a (natural) logarithmic scale the mean errors of the three datasets in Figures 5 to 7. For Gisette and random Gaussian datasets, using the learned sketches reduced the error by 10%–30%, and for the Swarm Behavior dataset, the learned sketches reduce the error by about 30%–40%.

## 5.3 Matrix Estimation with Nuclear Norm Constraint

In many applications, for the problem

$$X^* := \arg\min_{X \in \mathbb{R}^{d_1 \times d_2}} \|AX - B\|_F^2 \,,$$

it is reasonable to model the matrix $X^*$ as having low rank. Similar to $\ell_1$-minimization for compressive sensing, a standard relaxation of the rank constraint is to minimize the nuclear norm of $X$, defined as $\|X\|_* := \sum_{j=1}^{\min\{d_1, d_2\}} \sigma_j(X)$, where $\sigma_j(X)$ is the $j$-th largest singular value of $X$.

Hence, the matrix estimation problem we consider here is

$$X^* := \arg\min_{X \in \mathbb{R}^{d_1 \times d_2}} \|AX - B\|_F^2 \text{ such that } \|X\|_* \leq \rho.$$

where $\rho > 0$ is a user-defined radius as a regularization parameter.

---

[3]https://archive.ics.uci.edu/ml/datasets/Swarm+Behaviour
[4]https://www.csie.ntu.edu.tw/~cjlin/libsvmtools/datasets/binary.html#gisette

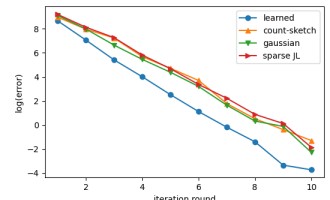

Figure 8: Test error of matrix estimation on synthetic data, $m = 50$

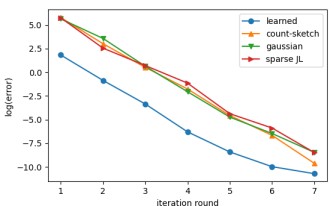

Figure 9: Test error of matrix estimation on Tunnel dataset, $m = 50$

We conduct experiments on the following two datasets:

- **Synthetic Dataset**: We generate the pair $(A_i, B_i)$ as $B_i = A_i X_i^* + W_i$, where $A_i \in \mathbb{R}^{n \times d_1}$ with i.i.d $N(0,1)$ entries. $X_i^* \in \mathbb{R}^{d_1 \times d_2}$ is a matrix with rank at most $r$. $W_i$ is noise with i.i.d $N(0, \sigma^2)$ entries. Here we set $n = 500, d_1 = d_2 = 7, r = 3, \rho = 30$. $|(A, B)|_{\text{train}} = 270, |(A, B)|_{\text{test}} = 30$.
- **Tunnel**[5]: The data set is a time series of gas concentrations measured by eight sensors in a wind tunnel. Each $(A, B)$ corresponds to a different data collection trial. $A_i \in \mathbb{R}^{13530 \times 5}, B_i \in \mathbb{R}^{13530 \times 6}, |(A, B)|_{\text{train}} = 144, |(A, B)|_{\text{test}} = 36$. The same dataset and parameters were also used in (Liu et al., 2020) for regression tasks. In our nuclear norm constraint, we set $\rho = 10$.

We choose $m = 40, 50$ for the synthetic dataset and $m = 10, 50$ for the Tunnel dataset and define the error to be $\frac{1}{2}(\|AX - B\|_F^2 - \|AX^* - B\|_F^2)$. For each data point, we take the average error of five independent trials. The mean errors of the two datasets when $m = 50$ are plotted in a (natural) logarithmic scale in Figures 8 and 9. We observe that the classical sketches yield approximately the same order of error, while the learned sketches improve the error by at least 30% for the synthetic dataset and surprisingly by at least 95% for the Tunnel dataset. The huge improvement on the Tunnel dataset may be due to the fact that the matrices $A_i$ have many duplicate rows. We defer the results when $m = 10$ or $40$ to Appendix D, which show that the learned sketches yield much smaller errors than the random sketches and the random sketches could converge significantly more slowly with considerably larger errors in the first several rounds.

## 5.4 RUNTIME OF LEARNED SKETCHES

As stated in Section 2, our learned sketch matrices $S$ are all COUNT-SKETCH-type matrices (each column contains a single nonzero entry), the matrix product $SA$ can thus be computed in $O(\text{nnz}(A))$ time and the overall algorithm is expected to be fast. To verify this, we plot in an error-versus-runtime plot for the SVM and matrix estimation with nuclear norm constraint tasks in Figures 10 and 11 (corresponding to the datasets in Figures 7 and 9). The runtime consists only of the time for sketching and solving the optimization problem and does not include the time for loading the data. We run the same experiment three times. Each time we take an average over all test data. From the plot we can observe that the learned sketch and COUNT-SKETCH have the fastest runtimes, which are slightly faster than that of the SJLT and significantly faster than that of the Gaussian sketch.

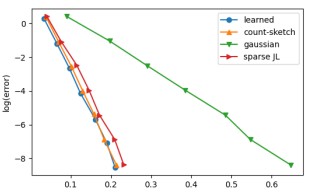

Figure 10: Test error of SVM on Gisette dataset

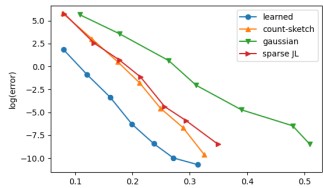

Figure 11: Test error of matrix estimation on Tunnel dataset

## 6 FAST REGRESSION EXPERIMENT

We consider the unconstrained least-squares problem, i.e., (5) with $\lambda = 0$, using the CO emission, greenhouse gas datasets, and the following Census dataset:

- Census data[6]: this dataset consists of annual salary and related features on people who reported that they worked 40 or more weeks in the previous year and worked 35 or more hours per week. We randomly sample 5000 instances to create $(A_i, b_i)$, where $A \in \mathbb{R}^{5000 \times 11}$ and $b \in \mathbb{R}^{5000 \times 1}$, $|(A, B)|_{\text{train}} = 160, |(A, B)|_{\text{test}} = 40$.

---

[5]https://archive.ics.uci.edu/ml/datasets/Gas+sensor+array+exposed+to+turbulent+gas+mixtures
[6]https://github.com/chocjy/randomized-quantile-regression-solvers/tree/master/matlab/data

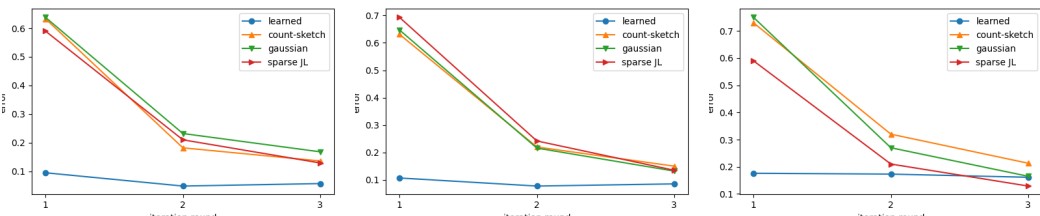

Figure 12: Test error of fast regression for the CO emission dataset, first three calls in solving an unconstrained least-squares

**Training.** We optimize the learned sketch $S_1$ by gradient descent (Algorithm 1), where $\mathcal{L}(S, A) = \kappa(AR_1^{-1})$, where $R_1$ is computed as in Algorithm 3 and $\kappa(M)$ denotes the condition number of a matrix $M$.

Next we discuss how to generate the training data. Since we use Newton's method to solve an unconstrained convex optimization problem (see Section 2), in the $t$-round, we need to solve a regression problem $\min_z \left\|(\nabla^2 f(x_t)^{1/2})^\top (\nabla^2 f(x_t)^{1/2})z - \nabla f(x_t)\right\|_2$. Reformulating it as $\min_z \left\|A^\top A z - y\right\|_2$, we see that $A$ and $y$ depend on the previous solution $x_t$. Hence, we take $x_t$ to be the solution obtained from Algorithm 3 using the learned sketch $S_t$, and this generates $A$ and $y$ for the $(t + 1)$-st round.

**Experiment.** For the CO emission dataset, we set $m = 70$, and for the Census dataset, we set $m = 500$. For the $\eta$ in the Algorithm 3, we set $\eta = 1$ for the first round and $\eta = 0.2$ for the next rounds for the CO emission dataset, and $\eta = 1$ for all rounds for the Census dataset. We leave the setting and results on the greenhouse gas dataset to Appendix E.

We examine the accuracy of the subproblem (4) and define the error to be $\left\|A^\top A R z_t - y\right\|_2 / \|y\|_2$. We run the first three subroutines of solving the subproblem for the CO emission dataset and the Census dataset. The average error of three independent trials is plotted in Figures 12, 13 and 16. We observe that for the CO emission dataset, the classical sketches have a similar performance and the learned sketches lead to a fast convergence in the subroutine with the first-round error at least 80% smaller; for the Census dataset, the learned sketch achieves the smallest error for all three rounds, where we reduce about 60% error in the first round and about 50% error in the third round. Note that the learned sketch always considerably outperforms COUNT-SKETCH in all cases.

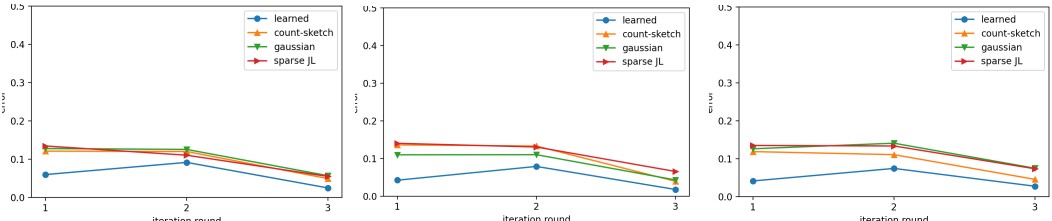

Figure 13: Test error of fast regression for the Census dataset, first three calls in solving an unconstrained least-squares

## 7 CONCLUSION

We demonstrated the superiority of using learned sketches, over classical random sketches, in the Iterative Hessian Sketching method and fast regression solvers for unconstrained least-squares. Compared with random sketches, our learned sketches of the same size can considerably reduce the error in the loss function (i.e., $f(x) - f(x^*)$, where $x$ is the output of the sketched algorithm and $x^*$ the optimal solution to the unsketched problem) for a given threshold of maximum number of iterations or maximum runtime. Learned sketches also admit a smaller sketch size. When the sketch size is small, the algorithm with random sketches may fail to converge, or converge slowly, while the algorithm with learned sketches converges quickly.

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

## A  PROOF OF LEMMA 3.1

Suppose that $AR^{-1} = UW$, where $U \in \mathbb{R}^{n \times d}$ has orthonormal columns, which form an orthonormal basis of the column space of $A$. Since $T$ is a subspace embedding of the column space of $A$ with probability 0.99, it holds for all $x \in \mathbb{R}^d$ that

$$\frac{1}{1+\eta} \left\| TAR^{-1}x \right\|_2 \le \left\| AR^{-1}x \right\|_2 \le \frac{1}{1-\eta} \left\| TAR^{-1}x \right\|_2 .$$

Since

$$\left\| TAR^{-1}x \right\|_2 = \left\| Qx \right\|_2 = \left\| x \right\|_2$$

and

$$\left\| Wx \right\|_2 = \left\| UWx \right\|_2 = \left\| AR^{-1}x \right\|_2 \tag{6}$$

we have that

$$\frac{1}{1+\eta}\left\|x\right\|_2 \le \left\|Wx\right\|_2 \le \frac{1}{1-\eta}\left\|x\right\|_2, \quad x \in \mathbb{R}^d. \tag{7}$$

It is easy to see that

$$Z_1(S) = \min_{x \in \mathbb{S}^{d-1}} \left\|SUx\right\|_2 = \min_{y \ne 0} \frac{\left\|SUWy\right\|_2}{\left\|Wy\right\|_2},$$

and thus,

$$\min_{y \ne 0}(1-\eta)\frac{\left\|SUWy\right\|_2}{\left\|y\right\|_2} \le Z_1(S) \le \min_{y \ne 0}(1+\eta)\frac{\left\|SUWy\right\|_2}{\left\|y\right\|_2}.$$

Recall that $SUW = SAR^{-1}$. We see that

$$(1-\eta)\sigma_{\min}(SAR^{-1}) \le Z_1(S) \le (1+\eta)\sigma_{\min}(SAR^{-1}).$$

By definition,

$$Z_2(S) = \left\|U^T(S^\top S - I_n)U\right\|_{op}.$$

It follows from (7) that

$$(1-\eta)^2\left\|W^T U^T(S^T S - I_n)UW\right\|_{op} \le Z_2(S) \le (1+\eta)^2\left\|W^T U^T(S^T S - I_n)UW\right\|_{op}.$$

and from (7), (6) and (Vershynin, 2012, Lemma 5.36) that

$$\left\|(AR^{-1})^\top(AR^{-1}) - I\right\|_{op} \le 3\eta.$$

Since

$$\left\|W^T U^T(S^T S - I_n)UW\right\|_{op} = \left\|(AR^{-1})^\top(S^T S - I_n)AR^{-1}\right\|_{op}$$

and

$$\left\|(AR^{-1})^\top S^T S AR^{-1} - I\right\|_{op} - \left\|(AR^{-1})^\top(AR^{-1}) - I\right\|_{op}$$
$$\le \left\|(AR^{-1})^\top(S^T S - I_n)AR^{-1}\right\|_{op}$$
$$\le \left\|(AR^{-1})^\top S^T S AR^{-1} - I\right\|_{op} + \left\|(AR^{-1})^\top(AR^{-1}) - I\right\|_{op},$$

it follows that

$$(1-\eta)^2\left\|(SAR^{-1})^\top SAR^{-1} - I\right\|_{op} - 3(1-\eta)^2\eta$$
$$\le Z_2(S)$$
$$\le (1+\eta)^2\left\|(SAR^{-1})^\top SAR^{-1} - I\right\|_{op} + 3(1+\eta)^2\eta.$$

We have so far proved the correctness of the approximation and we shall analyze the runtime below.

Since $S$ and $T$ are sparse, computing $SA$ and $TA$ takes $O(\mathrm{nnz}(A))$ time. The QR decomposition of $TA$, which is a matrix of size $\mathrm{poly}(d/\eta) \times d$, can be computed in $\mathrm{poly}(d/\eta)$ time. The matrix $SAR^{-1}$ can be computed in $\mathrm{poly}(d)$ time. Since it has size $\mathrm{poly}(d) \times d$, its smallest singular value can be computed in $\mathrm{poly}(d)$ time. To approximate $Z_2(S)$, we can use the power method to estimate $\left\|(SAR^{-1})^T SAR^{-1} - I\right\|_{op}$ up to a $(1 \pm \eta)$-factor in $O((\mathrm{nnz}(A) + \mathrm{poly}(d))\log(1/\eta))$ time.

## B  PROOF OF THEOREM 3.2

In Lemma 3.1, we have with probability at least 0.99 that

$$\frac{\widehat{Z}_2}{\widehat{Z}_1} \ge \frac{\frac{1}{(1+\eta)^2}Z_2(S) - 3\eta}{\frac{1}{1-\eta}Z_1(S)} \ge \frac{1-\eta}{(1+\eta)^2}\frac{Z_2(S)}{Z_1(S)} - \frac{3\eta}{Z_1(S)}.$$

When $S$ is random subspace embedding, it holds with probability at least 0.99 that $Z_1(S) \ge 3/4$ and so, by a union bound, it holds with probability at least 0.98 that

$$\frac{\widehat{Z}_2}{\widehat{Z}_1} \ge \frac{1}{(1+\eta)^4}\frac{Z_2(S)}{Z_1(S)} - 4\eta,$$

or,

$$\frac{Z_2(S)}{Z_1(S)} \le (1+\eta)^4 \left( \frac{\widehat{Z}_2}{\widehat{Z}_1} + 4\eta \right).$$

The correctness of our claim then follows from (Pilanci & Wainwright, 2016, Proposition 1), together with the fact that $S_2$ is a random subspace embedding. The runtime follows from Lemma 3.1 and (Cormode & Dickens, 2019, Theorem 2.2).

## C    PROOF OF THEOREM 4.1

The proof follows almost an identical argument as that in (van den Brand et al., 2020, Lemma B.1). In (van den Brand et al., 2020), it is assumed (in our notation) that $3/4 \le \sigma_{\min}(AP) \le \sigma_{\max}(AP) \le 5/4$ and thus one can set $\eta = 1$ in Algorithm 3 and achieve a linear convergence. The only difference is that here we estimate $\sigma_{\min}(AP)$ and $\sigma_{\max}(AP)$ and set the step size $\eta$ in the gradient descent algorithm accordingly. By standard bounds for gradient descent (see, e.g., (Boyd & Vandenberghe, 2004, p468)), with a choice of step size $\eta = 2/(\sigma_{\max}^2(AP) + \sigma_{\min}^2(AP))$, after $O((\sigma_{\max}(AP)/\sigma_{\min}(AP))^2 \log(1/\epsilon))$ iterations, we can find $z_t$ such that

$$\left\| P^\top A^\top AP(z_t - z^*) \right\|_2 \le \epsilon \left\| P^\top A^\top AP(z_0 - z^*) \right\|_2,$$

where $z^* = \arg\min_z \left\| P^\top A^\top APz - P^\top y \right\|_2$ is the optimal least-squares solution. This establishes Eq. (11) in the proof in (van den Brand et al., 2020), and the rest of the proof follows as in there.

## D    IHS EXPERIMENT: MATRIX ESTIMATION WITH NUCLEAR NORM CONSTRAINT

As stated in Section 5.3, the mean errors of the two datasets when $m = 40$ for the synthetic dataset and $m = 10$ for the Tunnel dataset are plotted in a (natural) logarithmic scale in Figures 14 and 15.

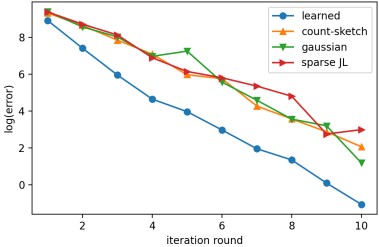
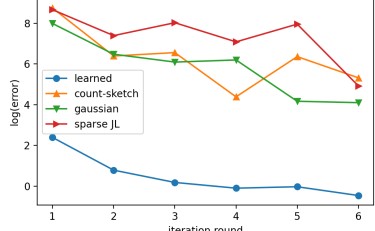

Figure 14: Test error of matrix estimation on synthetic dataset, $m = 40$

Figure 15: Test error of matrix estimation on Tunnel dataset, $m = 10$

## E    FAST REGRESSION EXPERIMENT: GREENHOUSE GAS

We set $m = 100$, $\eta = 0.2$ in the first round and $\eta = 1$ in the second round. The mean errors of the first two calls of the fast regression subroutine are plotted in Figure 16. We can observe that Gaussian and sparse JL sketches have a significantly better performance than COUNT-SKETCH, and the learned sketch shows again a significant reduction of more than 50% in the first-round error.

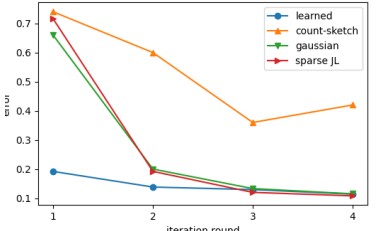 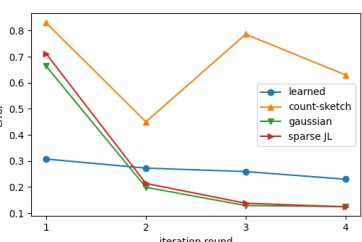

Figure 16: Test error of fast regression for the greenhouse gas emission dataset, first two calls in solving an unconstrained least-squares

