# OpenReview forum: "Learning-Augmented Sketches for Hessians"
_ICLR.cc/2021/Conference — Reject_

### Official Review · AnonReviewer2 · 2020-10-17
**An interesting approach for accelerated sketching-based regression solver, but the practicality seems unconvincing, and the numerical results could be misleading**

**Rating:** 4
**Confidence:** 5

**Review:**

The paper proposed a learned variant of the well-known iterative Hessian sketch (IHS) method of Pilanci and Wainwright, for efficiently solving least-squares regression. The proposed method is essentially a learned variant of the count-sketch, where the positions of the non-zero entries are random while the value is learned. While getting a learned variant for IHS is an interesting direction, the current theoretical contribution of this paper is only incremental, and most importantly, the reviewer is unconvinced for the practicality of the current approach.

The learned sketching matrix S^t is computed at each iteration, which should introduce a significant computational overhead. Then the plots comparing the learned IHS with unlearned IHS is unfair since it is regarding the iteration count. The iteration count does not reflect such overhead, and the reviewer believe that the wall-clock time comparision would be more sensible.

The reviewer found that the authors are over-selling their numerical results. For example in Figures 1, 2, 6, 7, that the learned IHS has the same linear convergence rate as the unlearn IHS, and actually only slightly faster overall (only 1 iteration faster if we view the plots horizontally), but the authors' claims are like “We can observe that the classical sketches have approximately the same order of error and the learned sketches reduce the error by a 5/6 to 7/8 factor in all iterations”, “We observe that the classical sketches yield approximately the same order of error, while the learned sketches improve the error by at least 30% for the synthetic dataset and surprisingly by at least 95% for the Tunnel dataset.”, which the authors seem to purposefully report from vertical views to exaggerate the results. This is very misleading to the readers which are not familiar with the related works.

********* Update ********

The reviewer appreciates the efforts authors have made. However, the response does not fully address the issues. In the new experiments the authors used small sketch size to show clearer advantages of the learned IHS (Figure 2 & 4), but at that regime all of the methods including learned-IHS are converging only slowly and not practical compare to the slightly larger sketch size choices. The reviewer believes that such a comparison is not meaningful. In order to truly demonstrate the benefits of learned-IHS (which seems to be robust to small sketch sizes), the authors should choose for each algorithm the best sketch size and then compare them in run time, at least between learned-IHS and Count-sketch IHS.

Beside the flaws in numerical experiments, the reviewer found that the theoretical contribution of the current version is incremental. From the reviewer's point of view, the meaningful analysis for learned IHS should certainly be how the converge relates to  the statistics of the training data, to show the benefit of the learned sketch theoretically (i.e. to show how much better the learned sketch SA compare to the unlearned ones in terms of Z_1 and Z_2). The authors have avoided such type of analysis and the main results are "safe-guarded" by the concentration of the random sketch, which are easy to derive.

Overall, the idea of combining IHS with learned sparse sketch is interesting, but the reviewer believes that the current version needs significant amount of rework to be publishable in top conferences.

---

> ### Author Response · Authors · 2020-11-18
> **Response to Reviewer 2**
>
> We thank the reviewer for his/her comments. We have made edits to the manuscript with changes marked in blue.
>
>     > The learned sketching matrix S^t is computed in each iteration, which should introduce a significant computational overhead. Then the plots comparing the learned IHS with unlearned IHS are unfair since it is regarding the iteration count. The iteration count does not reflect such overhead, and the reviewer believes that the wall-clock time comparison would be more sensible.
>
> The learned sketching matrix S^t is trained only once using the training data and can be completed offline. The sketch matrix S^t is not computed while solving the optimization problem on the test data. We just use the sketch matrix which has been trained while running the algorithm on the test data and there is no additional computational cost in generating S^t other than solving the iterative step using a random Count-Sketch matrix (in each round we compare the learned sketch with a random sketch and choose the better one).
>
> Regarding a wall-clock time comparison, since our learned sketch is of the Count-Sketch type (each column has only one non-zero entry), the total running time is almost the same as that of the Count-Sketch matrix and is faster than Gaussian and Sparse-JL matrices. We have added a plot to show this.
>
> Another reason we use iteration count is the following. If we use wall-clock time, it is clear that the learned sketch and Count-Sketch will run faster than sparse-JL and significantly faster than the Gaussian matrix (because the Gaussian matrix is dense). What is not clear is the error, e.g., whether the Gaussian matrix gives a smaller error. Hence, our goal is to compare the performance between Count-Sketch matrices and other types of matrices, and thus we choose to compare the error iteration by iteration.
>
>     > The learned IHS has the same linear convergence rate as the unlearned IHS, and is actually only slightly faster overall (only 1 iteration faster if we view the plots horizontally).
>
> The effect of 1 iteration faster is a coincidence. We have added two figures for the LASSO experiments with a smaller sketch size, which demonstrate that learned sketches can converge fast and yield a small error in a few iterations. Meanwhile random sketches may not converge, or give significantly larger error, even with  approximately the same number of iterations.

---

> > ### Author Response · Authors · 2020-11-24
> > **More experiments included**
> >
> > In addition to the two additional LASSO experiments mentioned in the earlier response, we have included two more experiments of matrix regression with smaller values of m (in Appendix D). The two new experiments again demonstrate that learned sketches can converge fast to a small error in a few iterations, while random sketches may not converge, or give significantly larger errors, even with approximately the same number of iterations.

---

### Official Review · AnonReviewer3 · 2020-10-23
**Review of Learning-Augmented Sketches for Hessians**

**Rating:** 6
**Confidence:** 4

**Review:**

SUMMARY:

Sketching is a popular technique in numerical linear algebra for achieving various desirable properties (e.g., lower complexity, one pass methods). The present paper considers a particular kind of sketch for which the sketch matrix is learned from data. It shows how such learned sketches can be used in two types of problems: Hessian sketching (Sec. 3) and Hessian regression (Sec. 4). The authors give both algorithms and provide theoretical guarantees. They also apply these techniques to a number of both synthetic and real datasets in the experiments. For the most part, the experiments indicate that the proposed methods give a consistent, but not necessarily very large, improvement.

I think the idea of using learned sketches is interesting, and it seems like it has not been applied to the problems considered in this paper before. I think the paper strikes a good balance between algorithms, theory and experiments. I like the paper, but there are a few issues that must be addressed before it can be published. Most importantly, there seems to be errors in both Lemma 3.1 and Theorem 3.2. These should hopefully be easy to fix though.

In the current state, I vote to reject the paper. But if my concerns, especially the issues with Lemma 3.1 and Theorem 3.2, can be addressed, I'll be happy to increase the rating.


ADVANTAGES:

- Learned sketching is a fairly new and interesting idea.
- It seems like learned sketching has not been used for Hessian sketching and regression before.
- Good balance of algorithms, theory and experiments.


CONCERNS/QUESTIONS:

- In the list that describes performance improvements under "Our Contributions", it is not clear what the quantities $x^*$ and $X^*$ represent. In Sections 5 and 6, these quantities are used to represent the optimal solution to the unsketched problems, but here they seem to represent the solution for the sketched problem using one of the random sketches. If it's the latter, are $x^*$ and $X^*$ the best performing solutions produced by the competing methods? It would be helpful if you clarified this.

- In Algorithm 1, how is $\alpha$ chosen? Is $\alpha$ updated adaptively, or is it just a fixed small number? Is there any rule of thumb for how to do this?

- Lemma 3.1: Is seems like the bound on $\hat{Z}_2$ stated in the lemma is wrong. Based on the upper and lower bounds on $Z_2(S)$ towards the end of Section A, it seems like this bound should read
$$
\frac{Z_2(S)}{(1+\eta)^2} - 3 \eta \leq \hat{Z}_2 \leq \frac{Z_2(S)}{(1-\eta)^2} + 3 \eta.
$$

- Lemma 3.1: Since you use that $\max(\eta,\eta^2) = \eta$ when applying the result of Vershynin (2012) in the proof, you should add the condition that $\eta \leq 1$ in the lemma statement.

- Proof of Lemma 3.1: In Section A, 2nd sentence, you say "Since $T$ is a subspace embedding of the column space of $A$...". I think you should add that this is true with probability at least 0.99, to clarify where the 0.99 probability of success in the lemma statement comes from.

- Proof of Lemma 3.1: In Section A, you use the inverse of $R$ in multiple places. However, it seems like $R$ won't be invertible unless $A$ is of full rank. Can the proof be adapted for a case when $A$ is rank deficient? If not, you should add that $A$ is assumed to be full rank in the lemma statement.

- Proof of Lemma 3.1: This is related to the previous point. In the second to last equation on page 10, it seems like the equation
$$
\min_{x \in S^{d-1}} || S U x || = \min_{y \neq 0} \frac{|| S U W y ||}{|| W y ||}
$$
only will hold if W is full rank, which requires A to be full rank (since $AR^{-1} = UW$ and $U$ has $d$ columns and is full rank). Can the proof be adapted for a case when $A$ is rank deficient? If not, you should add that $A$ is assumed to be full rank in the lemma statement.

- Theorem 3.2: Given the error in Lemma 3.1, the bound in Theorem 3.2 needs to be updated accordingly. Also, in the proof in Section B, in the first inequality you use
$$
\frac{1}{\hat{Z}_1} \geq \frac{1}{\frac{1}{1+\eta} Z_1(S)}.
$$
There's a sign error here; it should read
$$
\frac{1}{\hat{Z}_1} \geq \frac{1}{\frac{1}{1-\eta} Z_1(S)}.
$$

- Proof of Theorem 3.2: In Section B, 2nd sentence, it would be helpful for the reader if you said "holds with probability 0.99" instead of "holds with high probability", and then also clarify that you do a union bound with the 0.99 probability from Lemma 3.1 to get a success probability of at least 0.98 in Theorem 3.2. This may be obvious to readers familiar with the area, but being clear with these things would make the paper accessible to a wider audience.

- Solving (4) deterministically would cost $O(nd^2)$. For Alg. 3 to be worthwhile, it therefore seems like the number $\min(\sigma_1/\sigma_1', \sigma_2/\sigma_2')$ in Theorem 4.1 must be smaller than $d$. Could you say something about why we expect this $\min$ to be small? Is it the case that the $\min$ may be large with some small probability?

- In the experiments, $m$ is chosen as $m = kd$ for some integer $k$. This is different from the $m = O(d^2)$ required for CountSketch to be a subspace embedding. Have you found that this choice $m = O(d)$ always works well in practice, or have you encountered datasets where such a choice has proven to be too small?

- For the experiments in Section 5.2, is there any reason why you only use learned sketches in the first round for the Gaussian and Swarm behavior datasets, rather than use them all rounds as for the other datasets and experiments?

- In Section 6, is there a reason why you choose $\eta = 1$ and $\eta = 0.2$ rather than using the rule for setting $\eta$ in Alg. 3?

- In Section C, you refer to "standard bounds for gradient descent". Can you please provide a reference for those that are unfamiliar with the literature?

- The paper ends abruptly with no conclusion.


MINOR CONCERNS/QUESTIONS:

- On page 2, in the 2nd paragraph, 3rd sentence, you say "If $A$ is tall-and-skinny, then $S$ is wide-and-fat...". Should it be "short-and-fat" rather than "wide-and-fat"?

- The usage of R is a bit confusing. It is used to mean the R matrix in a QR factorization in Alg. 2, the inverse of the R matrix from a QR factorization in Alg. 3, and the upper bound on the nuclear norm in Section 5.3. It would be less confusing if a the inverses in Alg. 3 and the nuclear norm bounds were called something other than R.

- In Section 5.2, for the Swarm behavior and Gisette datasets, you say that $B_i$ is of size $2430 \times 30$ and $5030 \times 30$, respectively. Should this be $2400 \times 30$ and $5000 \times 30$ since $n$ is 2400 and 5000 respectively for the two datasets?

#######################

Update:

The authors have addressed my questions adequately. In particular, my main concerns with the theoretical results and proofs have been fixed. I have updated the score from 3 to 6 for now.

---

> ### Author Response · Authors · 2020-11-18
> **Response to Reviewer 3**
>
> We thank the reviewer for his/her careful review. We have made edits to the manuscript with changes marked in blue.
>
>     > Issues with Lemma 3.1 and Theorem 3.2
>
> We have corrected the proofs of Lemma 3.1 and Theorem 3.2, with the assumption that A is of full rank.
>
>     > it is not clear what the quantities x* and X* represent
>
> The quantities x* and X* do represent the optimal solution (of the unsketched problem) and are not the solution of the sketched algorithm. Here, we are comparing f(x) - f(x*) (where f is some loss function) for x returned by the algorithm using a learned sketch with f(x) - f(x*) for x returned by using a random sketch. In both cases, x* represents the optimal solution of the unsketched problem.
>
>     > In Algorithm 1, how is alpha chosen?
>
> We tune the learning rate alpha according to the performance on the training data. We use a fixed value for the training process, but can be different for different iterations in some tasks.
>
>     > Could you say something about why we expect this (min(sigma_1/sigma_1', sigma_2/sigma_2')) to be small? Is it the case that this may be large with some small probability?
>
> We have added a Remark 4.2 that the condition number is expected to be small when S_2 is a subspace embedding, and hence Algorithm 3 can be faster than the trivial O(nd^2). This fact has been used in previous regression solvers, see, e.g., [1, p38].
>
>     > In the experiments, m is chosen to be kd for some integer k. This is different from the required for CountSketch to be a subspace embedding.
>
> The theoretical guarantee for the CountSketch matrix is m = O(d^2/eps^2), but as a matter of experience, in practice often it still works when m is much smaller. This is because CountSketch is known to do better when the maximum leverage score is small, and when the maximum leverage score is at most 1/d, a linear in d dependence is possible (see, e.g., paragraph before Remark 6 in [2]). Note that such an assumption can be much weaker than that of uniform sampling, which requires the maximum leverage score to be at most d/n to be effective. Here we try m = kd and the experiments show this works well.
>
>     > For the experiments in Section 5.2, is there any reason why you only use learned sketches in the first round for the Gaussian and Swarm behavior datasets, rather than use them in all rounds as for the other datasets and experiments?
>
> In the SVM task, we experimented with using learned sketches throughout the test and found that they gave little improvement in subsequent rounds. Note that we can run the learned sketch and a random sketch in parallel in each iteration and choose the better result. We have updated the experiments to use learned sketches throughout.
>
>     > In Section 6, is there a reason why you choose eta rather than using the rule for setting it in Alg. 3?
>
> In the original algorithm of [3], eta is set to be 1. But in our dataset, we found that if n is not much larger than d, the sketch size is not large enough to preserve the singular values of B between [1/sqrt(2), sqrt(2)], which would make the iterations diverge. Hence, in order to make the algorithm converge, we made eta smaller (the eta in Alg. 1 is the theoretical bound). In practice, sometimes an eta that is larger than the theoretical result does not make the result worse. Hence, we tried a larger eta when possible.
>
>     > In Section C, you refer to "standard bounds for gradient descent". Can you please provide a reference for those that are unfamiliar with the literature?
>
> We have added a reference to the bounds for gradient descent.
>
>     > Should it be "short-and-fat" rather than "wide-and-fat"?
>
> Yes, it should be "short-and-fat". We have corrected it.
>
>     > The usage of R is a bit confusing ... It would be less confusing if the inverses in Alg. 3 and the nuclear norm bounds were called something other than R.
>
> We have changed the inverses in Alg. 3 to P and the norm bounds to rho.
>
>     > In Section 5.2, for the Swarm behavior and Gisette datasets, you say that B_i is of size 2430x30 and 5030x30, respectively. Should this be 2400x30 and 5000x30 since n is 2400 and 5000 respectively for the two datasets?
>
> As stated, B is the block matrix [(AD)^T, (1/sqrt(C)) I_d]^T, so the size of B is (n + d) x d.
>
>     > The paper ends abruptly with no conclusion.
>
> We have added a conclusion.
>
>
> [1] David P. Woodruff. Sketching as a tool for numerical linear algebra. 10(1–2):1–157, October 2014.
>
> [2] Bourgain, J., Dirksen, S. & Nelson, J. Toward a unified theory of sparse dimensionality reduction in Euclidean space. Geom. Funct. Anal. 25, 1009–1088 (2015).
>
> [3] Jan van den Brand, Binghui Peng, Zhao Song, Omri Weinstein. Training (Overparametrized) Neural Networks in Near-Linear Time. ITCS, 2021, to appear. (arXiv:2006.11648 [cs.LG])

---

> > ### Comment · AnonReviewer3 · 2020-11-21
> > **Thank you for clarifications**
> >
> > Thank you for these clarifications. I have updated my review.

---

### Official Review · AnonReviewer4 · 2020-10-28
**Official Blind Review #4**

**Rating:** 6
**Confidence:** 3

**Review:**

Summary:
Previous work has shown that sketching the Hessian can improve the running time of second-order optimization methods. However, these prior sketching techniques were data-oblivious. This paper leverages learned sketches to improve the convergence rate of second-order optimization methods.

Strengths:
+ The learned sketch generally converges better than the other baseline.

Weaknesses:
+ The technical novelty of the paper seems incremental. The learned sketch algorithm from (Liu et al., 2020) serves as a replacement for the data-oblivious sketch in these second-order optimization methods.

Questions:
1) In experiments, is it possible to plot the convergence rate of the non-sketched, full-sized Hessian? I would like to compare the best convergence rate against the other baselines.
2) What is the total running time of the learned sketch against the other baselines?
3) The paper asks whether we should apply the same learned sketch each iteration or train the sketch continuously? However, this question was not thoroughly investigated in the paper. In the experiments, the sketching matrix is learned in each iteration. Is there evidence that supports this choice?

References:
1) Simin Liu, Tianrui Liu, Ali Vakilian, Yulin Wan, and David P. Woodruff. On learned sketches for randomized numerical linear algebra. arXiv:2007.09890 [cs.LG], 2020. URL https://arxiv.org/ abs/2007.09890.

---

> ### Author Response · Authors · 2020-11-18
> **Response to Reviewer 4**
>
> We thank the reviewer for his/her comments. We have made edits to the manuscript with changes marked in blue.
>
>     > The convergence rate of the non-sketched, full-sized Hessian:
>
> The iterative Hessian sketch is for solving $x_{t+1} = \textrm{argmin}_x [ \frac{1}{2}\|SA(x-x_t)\|_2^2 - \langle A^T(b-Ax_t), x-x_t \rangle ]$. If $S$ is the identity matrix, i.e., the minimization is unsketched, then no matter what value $x_t$ takes, $x_{t+1}$ is the optimal solution to the original least squares problem $\min \|Ax-b\|_2$. It means that if we use the full Hessian, we will obtain the optimal solution immediately.
>
>     > The total running time compared to other baselines:
>
> Our learned sketch is based on Count-Sketch (each column has a single non-zero entry), so the total running time is almost the same as Count-Sketch and is faster than Gaussian and Sparse-JL matrices. We have added a plot to show this.
>
> Another reason we use iteration count is that if we compare the runtimes, the learned sketch and Count-Sketch matrices will be faster than sparse-JL and significantly faster than Gaussian matrices (because they are dense). But here we also want to see the performance comparison between Count-Sketch matrices and other types of sketching matrices, so we choose to use iteration count.
>
>     > Why do we use different sketching matrices for different rounds?
>
> We have tried the same sketch matrix for different iterations. The results show that even for random sketches (such as Count-Sketch, sparse JL, and Gaussian), using the same matrix for different rounds makes the error larger than using different sketching matrices. Sometimes the error in the ($t+1$)-st round is larger than that in the t-th round if the same matrix is used for these two rounds. Hence, different matrices for different rounds was observed to be a better choice. Also, if we learn $t$ sketching matrices simultaneously with gradient descent, it will be too slow and we would need to determine the value of $t$. Hence, we decided to use different sketching matrices in different rounds.

---

### Author Response · Authors · 2021-02-25
**Update on the Manuscript**

An updated version of this manuscript is available at https://arxiv.org/abs/2102.12317.

The updated version includes an additional improved bound on the subspace embedding dimension for the column space of a tall-and-thin matrix, using sparse matrices, under the assumption that the matrix has only a few rows of large leverage scores and the indices of such rows are known. The improvement is obtained by analyzing sparse sketches applied to flat vectors. The advantage of having an oracle to predict which rows have a large leverage score is also corroborated empirically.

---

### Decision · Program_Chairs · 2021-01-07
**Final Decision**

**Decision:**

Reject

**Comment:**

This paper considers convex optimization problems whose solutions involve the solution of linear systems defined in terms of the Hessian. It presents algorithms that reduce the runtime of standard iterative approaches to solving these problems by iteratively sketching the Hessian; the novelty lies in the fact that the authors use the idea of learned sketches which have been used prior for problems in data mining. In particular, the authors use the approach to learning sketches of Liu et al., 2020 to learn the entries in sparse sketching matrices for the Hessian, and propose using the Iterative Hessian Sketch algorithms of Pilanci and Wainright, 2016 to iteratively solve the concerned optimization problem. The advantages of learned sketches are that they may allow using smaller sketch sizes while making progress on the problem, as they are learned to work well on the distribution of Hessians from which the problem instance is drawn.

The consensus of the reviews is that the idea of using learned sketches for convex optimization seems to be novel, and this paper is an interesting attempt, but falls short of the level of contribution required for publication in ICLR. The main concern is that the theory provided for the use of the learned sketches is incremental: the analysis does not reflect the fact that the sketches are learned; instead, the algorithm builds in a safeguard by using both a random sketch and a learned sketch, and the analysis uses the properties of the random sketch to proceed. The empirical results are suggestive, but the convergence rates of the learned and random sketches do not vary much, so the benefits seem marginal for most of the problems considered (with the exception of a standard least squares problem, for which we know learned sketching performs well).

 The paper is recommended to be rejected, as the theory is weak, and the empirical results are borderline.